# Comparison between Ultrasonographic-Guided Temporal and Coronoid Approaches for Trigeminal Nerve Block in Dogs: A Cadaveric Study

**DOI:** 10.3390/ani14111643

**Published:** 2024-05-31

**Authors:** Álvaro Jesús Gutiérrez Bautista, Manon Mikic, Pablo E. Otero, Virginia Rega, Francisco Medina-Bautista, José Ignacio Redondo, Sabine Kästner, Adriano Wang-Leandro

**Affiliations:** 1Department of Small Animal Medicine and Surgery, University of Veterinary Medicine Hannover Foundation, Germany, Foundation, Bünteweg 9, 30559 Hannover, Germany; gutierrezbautistaalvarojesus@gmail.com (Á.J.G.B.);; 2Department of Anesthesiology and Pain Management, Facultad de Ciencias Veterinarias, Universidad de Buenos Aires, Buenos Aires C1427CWO, Argentina; 3Royal (Dick) School of Veterinary Studies, The University of Edinburgh, Easter Bush Veterinary Centre, Roslin, Midlothian EH25 9RG, UK; 4Animal Medicine and Surgery Department, University of Córdoba, Campus Universitario de Rabanales, Ctra. Madrid-Cádiz Km. 396, 14071 Córdoba, Spain; 5Departamento de Medicina y Cirugía Animal, Facultad de Veterinaria, Universidad Cardenal Herrera-CEU, CEU Universities, 46115 Valencia, Spain

**Keywords:** mandibular, maxillary, ophthalmic nerves, regional anaesthesia, canine, orbitotomy, analgesia, computed tomography

## Abstract

**Simple Summary:**

Periorbital surgeries are painful procedures that require effective pain management, and regional anaesthesia is an essential tool used to achieve it. To desensitise the area of the head involved, blockade of the branches of the trigeminal nerve (ophthalmic, maxillary, and mandibular nerves) is warranted. This technique is well described and employed in human medicine, but the literature on veterinary medicine (dogs) is scarce. This study aims to assess and compare two ultrasound-guided approaches for trigeminal nerve block. Thirteen dog heads were utilised, and following a preliminary anatomical assessment, procedures were conducted using temporal and coronoid approaches. The needle was advanced under ultrasonographic guidance from the dorsal aspect of the temporal area or ventral to the zygomatic arch, respectively. A computed tomography scan was performed with the needles in place and repeated after injection of a contrast medium/tissue dye mixture. Dissection of the heads was immediately performed thereafter. Needle position, contrast distribution, and nerve staining were evaluated and compared between the two techniques. Results indicate no significant difference between both techniques. Both methods demonstrate adequate distribution, with minimal intracranial spread of the injectate. Both techniques are promising, although further studies in live animals are required.

**Abstract:**

The trigeminal nerve is responsible for innervating the periorbita. Ultrasound-guided trigeminal block is employed in humans for trigeminal neuralgia or periorbital surgery. There are no studies evaluating this block in dogs. This study aims to evaluate and compare two approaches (coronoid and temporal) of the trigeminal nerve block. We hypothesised superior staining with the coronoid approach. Thirteen dog heads were used. After a preliminary anatomical study, two ultrasound-guided injections per head (right and left, coronoid and temporal approach, randomly assigned), with an injectate volume of 0.15 mL cm^−1^ of cranial length, were performed (iodinated contrast and tissue dye mixture). The ultrasound probe was placed over the temporal region, visualising the pterygopalatine fossa. For the temporal approach, the needle was advanced from the medial aspect of the temporal region in a dorsoventral direction. For the coronoid approach, it was advanced ventral to the zygomatic arch in a lateromedial direction. CT scans and dissections were conducted to assess and compare the position of the needle, the spread of the injectate, and nerve staining. No significant differences were found. Both approaches demonstrated the effective interfascial distribution of the injectate, with some minimal intracranial spread. Although the coronoid approach did not yield superior staining as hypothesised, it presents a viable alternative to the temporal approach. Studies in live animals are warranted to evaluate clinical efficacy and safety.

## 1. Introduction

Surgery in the region of the periorbita is indicated in cases such as periorbital tumours, zygomatic adenectomy, or removal of persistent or recurrent retrobulbar abscess [1,2]. The most common surgical approach is a lateral orbitotomy, with or without transection of the orbital ligament and zygomatic osteotomy or ostectomy [3]. In some cases, a transverse osteotomy of the vertical ramus of the mandible is also warranted to achieve better surgical exposure [4]. Orbitotomy is a painful surgical intervention, so an effective multimodal analgesic plan is mandatory [5]. Opioids commonly form the basis for perioperative pain management and have proven pre-emptive benefit [6], although side effects like cardiorespiratory depression, nausea, or dysphoria can occur [7]. Regional anaesthesia has revolutionised medical and surgical practice by providing a safe and effective alternative to general anaesthesia in numerous procedures [8]. It can reduce opioid consumption during intra- and postoperative periods [9,10]. Efficient perioperative pain relief is essential for realising enhanced recovery strategies. Perioperative pain management using additional peripheral nerve blocks has been reported to ensure early extubation and shorter intensive care and hospitalisation times [11]. 

The innervation of the anatomical structures involving the periorbita corresponds to branches of the fifth cranial nerve, the trigeminal nerve [12], which is composed of motor and sensory fibres, with its nucleus located in the pons. It courses through the trigeminal canal (a bony furrow on the inside of the caudal part of the cranial fossa near the tympanic bulla) before dividing into three branches. The ophthalmic branch, or V_1_, exits the skull through the orbital fissure, passing through the peribulbar space, where it further divides into the frontal, lacrimal, and nasociliary nerves, providing sensory innervation to the upper eyelid, lateral orbit, medial conjunctiva, and globe. The maxillary branch, or V_2_, exits the skull through the foramen *rotundum* to the alar canal and enters the orbita through the rostral alar foramen, passing through the peribulbar space. Then, it divides into the zygomaticofacial and zygomaticotemporal nerves, providing sensory innervation to the upper and lower eyelids, conjunctiva, hard and soft palates, upper dental arch, and the rostral and maxillary aspect of the face. The mandibular branch, or V_3_, emerging to the orbit through the foramen ovale, provides sensory innervation to the buccal mucosa, mandibular teeth, and the skin below the mouth and motor innervation to the masseter, the temporalis, the pterygoids, the anterior belly of the digastric, and the mylohyoid muscles [13,14,15,16].

In human medicine, the trigeminal nerve block is commonly used to manage chronic pain conditions such as trigeminal neuralgia [17] and provide perioperative analgesia in facial surgery. In a randomised, prospective, double-blinded study, patients that were undergoing maxillofacial surgery required less intraoperative and postoperative opioid consumption when a preoperative trigeminal block was performed [18]. The coronoid is the most commonly ultrasound-guided approach performed, positioning the ultrasound probe underneath the zygomatic arch [19]. However, an approach above the zygomatic arch has also been reported, blocking the mandibular and maxillary nerves but proving ineffective for the ophthalmic branch [20].

In veterinary medicine, the temporal approach for blocking the trigeminal nerve was first described by Otero and Portela [21], and it was reported as part of the anaesthesia protocol in a case report detailing successful perioperative pain management in a dog undergoing a head mass excision using this approach [22]. However, to the authors’ knowledge, no anatomical cadaveric studies evaluating this technique or injection volume have been conducted in dogs.

The objectives of the present investigation are twofold: first, to describe the technique and feasibility of a coronoid approach for the trigeminal block in dogs; second, to evaluate and compare two different ultrasonographic-guided techniques for trigeminal nerve block, the temporal and coronoid approaches. We hypothesised that the coronoid approach would achieve a better distribution and nerve staining degree than the temporal approach.

## 2. Materials and Methods

This prospective, randomised, double-blinded, comparative, experimental study enrolled 14 mesocephalic dogs from different breeds with no head abnormalities. All animals died or were euthanised for reasons unrelated to the study. The heads were separated from the rest of the body and stored frozen. They were thawed for 24 h at room temperature before the date the study was conducted. During the preliminary phase, one of the heads was dissected to evaluate the anatomy, boundaries, and approaches to be compared. In the study phase, 13 heads were used to evaluate the needle trajectory and to compare the injectate spread and nerve staining. These characteristics were evaluated and documented by means of CT scans and subsequent anatomical dissections. The orbital fissure was defined as the target region as it represents the middle point of the sphenoidal complex and is where most of the nerves responsible for somatic innervation of the maxilla and the periocular tissues emerge from the skull. All the heads received bilateral injections in a randomised manner (www.random.org, accessed on 5 June 2023), with one side receiving the temporal and the other receiving the coronoid approach. Opaque envelopes were used to conceal allocation, ensuring only the treating investigator knew the assignment.

### 2.1. Preliminary Phase: Anatomical Study

The head of a mixed-breed, 12 kg male dog was used for anatomical investigation. Quincke spinal needles 22G (BD^®^ Spinal needle 0.7 × 75 mm, Madrid, Spain) were used to perform both approaches as described further in this paper, and a preliminary computed tomography was performed. Afterwards, an incision was made on the skin at the midpoint between the cranial border of the base of the ear and the lateral canthus of the eye, in a dorsoventral direction, from the dorsal midline of the skull to the ventral border of the mandible. The skin was dissected and removed, exposing temporal and masseters muscles, zygomatic arch, and temporomandibular joint. An oscillating saw was used to make two cuts at the zygomatic arch’s most caudal and most rostral aspects. The bone was then removed along with most of the masseter muscle. Two additional cuts were made during the coronoid process, and the middle part of the body of the mandible and the temporomandibular joint was luxated to expose the medial aspect of the mandibular ramus, where the mandibular foramen with the mandibular nerve (caudal alveolar nerve) were identified. Subsequently, the temporal muscle was incised and carefully dissected from the parietal and temporal bones in a ventromedial direction until the optic canal, the orbital fissure, and the rostral alar foramen were identified in the sphenoid bone. The caudal alveolar nerve was followed to its origin: the mandibular nerve in the foramen *ovale*. The mandibular, maxillary, and bundle of nerves emerging from the orbital fissure, corresponding to the oculomotor, trochlear, abducens, and ophthalmic nerves, were identified (Appendix A).

### 2.2. Study Phase: Ultrasound-Guided Trigeminal Nerve Block Injection

Considering an α of 0.05, a β of 0.1, and a significant difference in two nerve branches stained, the sample size required was five injections per group (a total of five animals, considering bilateral injection). Thirteen thawed mesocephalic canine heads from different breeds were used to mitigate a type 2 statistical error in case of possible data loss and to increase the power of the study. The volume to be injected was based on cranial length (CrL, cranial length is the distance from inion to nasion in centimetres (cm); Appendix A), extrapolated from Foster et al. in 2021 [23]. They reported that a volume of 0.2 mL cm^−1^ of cranial length injected in the peribulbar space in dogs resulted in a 50% intracranial spread. We decided to reduce the volume to 0.15 mL cm^−1^ of CrL to reduce the frequency of intracranial spread. The injectate consisted of a mixture of 1/5 iodinated contrast media (Imeron^®^ 300 mg/mL, Bracco Imaging Deutschland, GmbH, Konstanz, Germany) + 4/5 of blue dye (Tissue Marking Dyes^®^, Blue. Cancer Diagnostics, Inc, Durham, NC, USA.). The original dye was diluted 1:1000 in 0.9% NaCl solution.

The area caudal to the orbital ligament was clipped and cleaned to prepare the heads for the injections. The heads were positioned like a dog would be positioned in lateral recumbency, with the side to be injected uppermost. Alcohol 70% was applied to the skin to improve the acoustic coupling with the ultrasound probe. A portable ultrasound machine (Samsung^®^ HM70EVO; Samsung Electronics GmBH, Schwalbach, Germany) with a microconvex probe (4–10 MHz) was used for the procedure. The probe was covered with a dedicated ultrasound latex cover, and ultrasound gel was used between both to improve the coupling and image quality. Quincke spinal needles 22G (BD^®^ Spinal needle 0.7 × 75 mm, Madrid, Spain) were used to perform the approaches.

The ultrasound probe was positioned over the temporal region immediately caudal to the orbital ligament to obtain a transverse view of the caudal portion of the pterygopalatine fossa for both approaches. Slight tilting movements adjusted the probe until the following structures were identified: the frontal bone was observed as a hyperechogenic continuous line, which in the deep aspect of the pterygopalatine fossa showed an irregular structure consistent with the sphenoidal complex (optic canal, orbital fissure, and rostral alar canal). The coronoid process was also identified as a sharp, small, strongly reflective, hyperechogenic structure casting a strong acoustic shadow through the fossa (Figure 1 and Figure 2).

Description of the temporal approach: the needle was inserted using an in-plane approach from the medial aspect of the temporal region in a ventral direction until its tip was located close to the sphenoidal complex (Figure 1), where the calculated volume was injected.

Description of coronoid approach: the dog’s mouth was opened with a mouth gag to expose the mandibular notch and facilitate the insertion of the needle from lateral to medial, just ventral to the zygomatic arch and cranial to the temporomandibular joint (Figure 2). The needle was inserted using an in-plane approach, from lateral to medial direction until its tip was positioned near the sphenoidal complex, where the calculated volume was injected.

### 2.3. Computed Tomography

A computed tomography (CT) scan of the head was performed to document the needle position and distribution of the injectate. Image acquisition used a spectral detector CT scanner (Philips IQon Spectral CT, Philips Healthcare, Böblingen, Germany). The CT images were acquired with a slice thickness of 0.8 mm, helical pitch of 0.39, tube current of 430 mAs, tube potential of 120 kVp, matrix 512 × 512, and soft tissue and bone window reconstruction algorithm. Images were reviewed using a bone window (window level 800, window width 2000). Two scans were performed per head: the first one after placing the needles; the second after injecting the dye/contrast solution and removal of the needles. Evaluators performing the injections were blinded to the first CT scan findings.

The following CT features were recorded: distance from the tip of the needle to the target point in millimetres (mm), intracranial presence of contrast, intraconal presence of contrast, intramuscular presence of contrast, interfascial distribution of contrast and presence of contrast in the *foramina* of interest (optic canal, orbital fissure, rostral alar foramen, and foramen *ovale*). All measurements were performed by a single evaluator (MM). Except for the evaluation of the needle position, the evaluator was blinded to the approach performed for all the other parameters.

### 2.4. Dissections

Immediately after the CT scans were performed, heads were transported to a dedicated anatomy room and dissected by a researcher (AG) blinded to the approach performed. The skin between the lateral canthus of the eye and the cranial border of the ear canal was incised and removed. The zygomatic arch was cut as close as possible to the temporal and maxillary bones and removed. The coronoid process was also cut and removed from the mandible. The body of the mandible was cut to turn outwards and identify the mandibular foramen and the caudal alveolar nerve. The temporal muscle was carefully dissected and separated from the parietal and frontal bones ventromedially until the optic canal, orbital fissure, and rostral alar foramen were identified. The caudal alveolar nerve was followed to its origin (mandibular nerve in the oval foramen). The mandibular nerve, maxillary nerve, bundle of nerves emerging from the orbital fissure (oculomotor, trochlear, abducens and ophthalmic nerves), and the optic nerve were identified, and staining was recorded. Due to the complexity of differentiating individually the nerves emerging from the orbital fissure, where the ophthalmic nerve is located, the bundle was evaluated as a unit (i.e., orbital fissure complex).

While the optic canal and optic nerve were not directly targeted in the trigeminal block, they were both examined owing to the significance of this nerve, its proximity to other structures of interest and possible side effects if blocked.

### 2.5. Statistical Analysis

Statistical analysis was conducted using R version 4.3.2 and comprised two stages. Firstly, a descriptive analysis characterised the distribution of key variables. Continuous data were presented as median [range], while categorical data were expressed as frequency tables with the number and percentage of observations. Secondly, the study compared the distance from the tip of the needle to the target between approaches. The normality of the data was assessed using the Shapiro–Wilk test, which showed that the distance variable had a non-normal distribution. Consequently, a one-way trimmed means test was performed using the t1 way function from the WRS2 package in R. The Fisher’s test was employed to investigate possible associations between categorical variables. Statistical significance in all analyses was determined at *p* < 0.05.

## 3. Results

The preliminary study showed that in the coronoid approach if the needle advanced as far as the sphenoidal complex, it could be inserted inside the orbital fissure and therefore considered intracranial. Three heads had to be excluded due to anatomical alterations during the freezing process, which altered the soft tissues, making it impossible to perform the technique correctly. Ten heads were included in the analysis. Some contrast was spread through the needle’s pathway after withdrawal in all heads; hence, the approach employed could be devised during CT image evaluation.

### Computed Tomography and Anatomical Dissection

No significant differences in the distance from the tip of the needle to the target point were observed between both approaches. The median [range] of the coronoid group was 8.9 [3–17.6] mm, while in the temporal group was 10.3 [4.4–34.7] mm (Figure 3).

There were no differences between groups in contrast distribution analysed by CT (Table 1, Appendix A) or anatomical dissection (Table 2, Appendix A). In both approaches, intracranial distribution occurred in 1 out of 10 injections. Intraconal contrast was detected only in one injection out of ten in the temporal approach group. Intramuscular contrast was detected in most injections performed in both groups, and interfascial distribution was evidenced in all injections. The contrast reached the optic canal, orbital fissure, rostral alar canal, and oval foramen in 4, 4, 4, and 3 out of 10 injections with the coronoid approach and 0, 2, 1 and 1 out of 10 injections with the temporal approach.

Results from dye distribution assessed by dissection are shown in Table 2 and Appendix A. Dye was visible intraconally, intramuscularly, and interfascially in 1, 4, and 8 out of 10 injections with the temporal approach and in 0, 3, and 10 out of 10 injections with the coronoid approach. The dye stained the optic nerve, orbital fissure complex, maxillary nerve, and mandibular nerve in 0, 3, 4, and 5 out of 10 injections with the temporal approach and in 1, 8, 8, and 5 out of 10 injections with the coronoid approach.

## 4. Discussion

This is the first study to evaluate trigeminal block and compare two approaches in canine cadavers. The results of this study suggest that the coronoid approach is not superior to the temporal approach; however, it is non-inferior. Consequently, the coronoid approach is a viable alternative to the temporal approach. Although there was no significant difference between the two approaches, it is noteworthy that the trigeminal block, as described, did not stain all three branches of the trigeminal nerve in a considerable number of specimens. This observation may also have implications for targeted analgesia, depending on which branches are effectively blocked. Nevertheless, it should be noted that the distribution in living animals may deviate from the outcomes observed in this study. Further studies and clinical trials are warranted to demonstrate its efficacy. Equivalent contrast spread was observed regarding intracranial, intraconal, interfascial, and intramuscular distribution. The incidence of intracranial contamination was low (1/10) in both approaches; however, it should be noted that contrast spread in cadaveric studies may not precisely correspond to the spread of injectable solution in live animals [24]. The position of the needle in the coronoid approach was not significantly closer to the target point than in the temporal approach. Nevertheless, as observed in the preliminary study, it is essential to carefully introduce the needle and cease its advancement before reaching the sphenoidal complex when using the coronoid approach. Otherwise, there is a risk of inadvertently inserting it through one of the *foramina* where nerves exit or even into the interior of the skull.

The CT scan results do not entirely align with those of the dissection. This discrepancy can be attributed to several factors. Firstly, the CT scan may reveal small amounts of contrast and subtle enhancements that could be overlooked during dissection. For instance, minimal contrast within the muscle may not be readily apparent during dissection but can be visualised in CT scan assessments. Secondly, manipulation during dissection could alter the anatomy and distribution of dye, potentially leading to misinterpretation. The presence of contrast in the foramina and intracranial regions is more readily discernible with CT scan imaging. Therefore, we concluded that cranial opening was unnecessary. Moreover, identifying individual nerve staining away from the foramina is more straightforward during dissection than via CT scan imaging. Hence, the combination of both, CT scan and dissection, was advantageous to understand the distribution of the dye/contrast solution and involved structures.

Trigeminal nerve block procedures in humans differ from the approach described in our study. In the so-called coronoid approach, the ultrasound probe is positioned just below the zygomatic arch in the axial plane, allowing for the visualisation of the coronoid and condylar processes, with the mandibular notch situated between them, providing a view of the deeper structures of the pterygopalatine fossa [25]. This is in contrast to our method, where the probe is placed above the zygomatic arch in a transverse position and caudally to the orbital ligament, following the description by Otero and Portela [21]. In the present study, the mandibular notch is the point through which the needle is advanced, towards the sphenoidal complex in the coronoid approach. The coronoid approach in human medicine is described as an out-of-plane technique due to the steep angle at which the needle is advanced, with its corresponding disadvantages, unlike our study where the needle is always kept in-plane with both approaches. In 2020, Zou et al. described a novel approach for humans above the zygomatic arch. In this case, the authors placed the ultrasound probe above the zygomatic arch in a longitudinal position, allowing the needle to be inserted in-plane [20]. In contrast to the present study, they placed the tip of the needle onto the surface of the maxilla once the temporal and pterygoid muscles were passed through, rather than in the vicinity of the sphenoidal complex. They achieved hypoesthesia of the areas innervated by the maxillary and mandibular nerves, while the skin innervated by the ophthalmic branch was not desensitised. This differs from our results, where the main nerves stained were the maxillary and ophthalmic, with the mandibular nerve stained less frequently. This difference may be attributed to anatomical distinctions between the two species. The location of the three branches of the trigeminal nerve and relationships within the skull in dogs may vary compared to humans. Additionally, the size and orientation of the *foramina* where they exit from the skull in relation to surrounding structures differ between species.

To our knowledge, the only reference in veterinary medicine regarding trigeminal nerve block is a case report of a dog undergoing exenteration and excision of the zygomatic arch with partial maxillectomy due to a mass. In that report, a trigeminal nerve block was performed using a temporal approach as part of balanced anaesthesia, and the dog did not require rescue analgesia during the procedure or in the first postoperative hours [22]. In contrast to our method, they utilised the pulsatile maxillary artery as a landmark for injection. Maxillary artery visualisation for further guiding of the needle placement can be advantageous when performing this block. The authors reported that the mandibular branch was anaesthetised. Our findings revealed staining of the mandibular branch in half of the cases using both approaches. Nonetheless, as previously mentioned, the spread of injectate may differ between cadaveric studies and live patients.

Several publications assessing ultrasound-guided peribulbar blocks showed results similar to our study. In one study, an ultrasound-guided peribulbar injection in dog cadavers was examined. The researchers positioned the ultrasound probe over the surface of the cornea. They inserted a Tuohy needle using a subzygomatic approach through the masseter muscle, posterior to the maxillary bone and anterior to the vertical ramus of the mandible [26]. The contrast agent was distributed in the periorbital extraconal space, similar to that observed in the present study. The authors also observed a higher frequency of contrast enhancement in the CT scan of the orbital fissure and rostral alar foramen compared to our study, with contrast even observed in the mandibular foramen. Additionally, intracranial contrast was detected in all cases, whereas in our study, it was only observed in 2 out of 20 injections. Variations in the injection site may explain these differences. The lateral aspect of the peribulbar (extraconal) space may be less restricted due to the absence of a bony landmark compared to the vicinity of the sphenoidal complex, where the frontal and sphenoid bones form the medial boundary of the peribulbar space. Consequently, further spread of contrast may be achieved with the peribulbar approach. Another possible explanation is that Mahler et al. did not perform the CT scan immediately after the injection, allowing for additional contrast distribution over time. Mahler et al. determined the volume of injectate relative to the animals’ body weights in kilograms, rendering direct comparison impractical with the volumes utilised in our study. Although effective, the technique described by this author involved an out-of-plane approach, which may limit the visualisation of the complete length and tip of the needle during the procedure, with potential associated complications. Foster et al. evaluated a peribulbar injection using a very similar approach to the temporal approach used in ours [23]. The ultrasound probe was positioned on the temporal region caudally to the orbital ligament, albeit unlike our study, in a longitudinal orientation rather than transverse. They inserted the needle in a dorsoventral direction, similar to the temporal approach. A higher volume of injectate was utilised compared to our study (0.2 mL cm^−1^ of CrL compared to 0.15 mL cm^−1^ of CrL) and administered in the dorsal aspect of the periorbita. Computed tomography scans revealed contrast spread reaching the rostral alar foramen, orbital fissure, and oval foramen, mirroring our findings. Intracranial spread of contrast was more frequent in this study compared to our results (5 out of 10 injections versus 2 out of 20 injections). No contrast was observed in the optic canal, in contrast to our findings, which revealed contrast in the optic canal in 4 out of 20 injections. Optic nerve blockade must be considered when performing the approach described in the present study. This discrepancy could stem from the increased volume of injectate employed, variances in injectate viscosity, and, as previously mentioned, the injection site. Central nervous system toxicity with local anaesthetics has been documented in both human [27] and veterinary medicine [28], highlighting the need for caution when selecting injectate volume and the approach for periorbital pterygopalatine fossa injections.

This study has several limitations. Firstly, results from a cadaveric study cannot be entirely extrapolated to a clinical setting for various reasons: complications such as haemorrhage cannot be evaluated, and the spread of contrast can differ in live animals due to variations in the viscosity of the injectate compared to the local anaesthetics employed, as well as other factors such as muscle tone, blood supply, lymphatic drainage, temperature, and time [29]. In human literature, complications or side effects of blocks in the pterygopalatine fossa are not frequently reported [17]; however, due to the proximity of the surrounding neurovascular structures, facial paralysis, difficulty chewing, diplopia, exophthalmos, oedema, persistent numbness of the face, and haematoma can occur [30]. Furthermore, in live animals, the spread of local anaesthetics into muscles and fasciae can provide analgesia by desensitising free nerve endings or nociceptors present in those structures [29]. Fascial plane blocks have demonstrated multiple sites of action, which may result in a wider area of analgesia than that corresponding to the nerves blocked. Secondly, contrast spread alongside the needle pathway was detected in all cases during CT scan visualisation. Although the observer interpreting the images was blinded, identification of the approach could be inferred by the needle contrast pathway. Thirdly, the challenging dissection of the pterygopalatine fossa required sawing of bone structures and invasive manipulation of the complex anatomical area; thus, further tissue staining and damage to fragile structures were possible.

## 5. Conclusions

This study suggests that the coronoid approach of the trigeminal nerve block exhibits a similar contrast distribution to the temporal approach and shows promise in terms of nerve staining. It serves as an alternative when the temporal approach cannot be performed and may offer effective analgesia in periocular surgeries, such as exenteration or zygomatic ostectomies. Further studies are necessary to evaluate different volumes of injectate. Despite the existence of a clinical report in veterinary medicine, controlled in vivo studies are warranted to assess the clinical efficacy and potential side effects in living animals.

## Figures and Tables

**Figure 1 animals-14-01643-f001:**
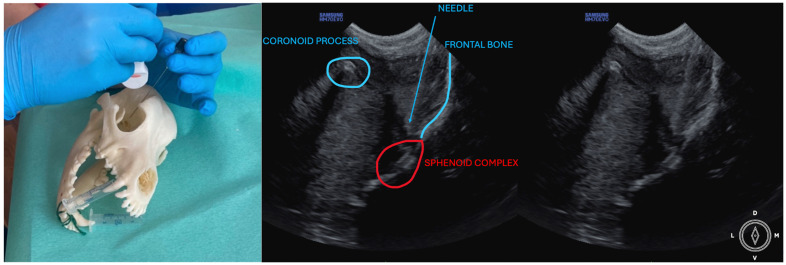
Position of the probe and needle. Ultrasound image of the temporal approach, where the anatomic landmarks can be identified. The microconvex ultrasound probe is placed on the temporal region, caudally to the orbital ligament, in a transversal position.

**Figure 2 animals-14-01643-f002:**
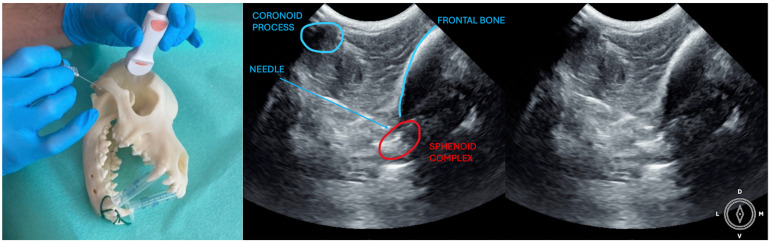
Position of the probe and the needle. Ultrasound image of the coronoid approach.

**Figure 3 animals-14-01643-f003:**
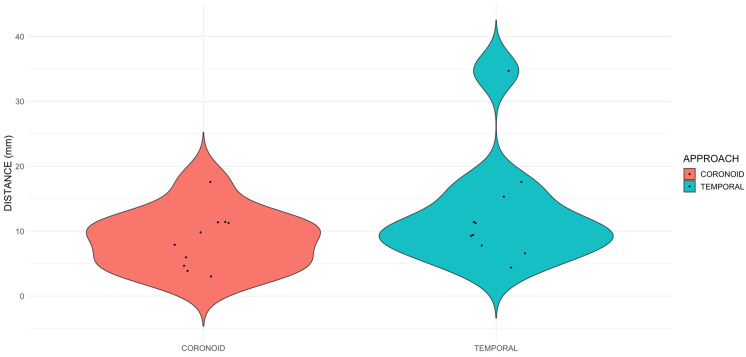
Violin plot of the distribution distance (mm) in the two approaches. The plot represents the distribution of the distances from the tip of the needle to the target point in millimetres in both approaches.

**Table 1 animals-14-01643-t001:** CT Study in ten specimens. Differences between both approaches in terms of the presence of hyperattenuating contrast (yes/no) within the structure (cranium, retrobulbar conus, muscle, and interfascial) or in contact with it (yes/no) (optic canal, orbital fissure, rostral alar foramen, and foramen ovale). Results are shown as a total frequency (n of 10) and as a percentage.

Approach	Coronoid				Temporal				*p*-Value
Contrast presence	YES	%	NO	%	YES	%	NO	%	
Intracranial	1	10	9	90	1	10	9	90	1
Intraconal	1	10	9	90	0	0	10	100	1
Intramuscular	10	100	0	0	10	100	0	0	1
Interfascial	10	100	0	0	10	100	0	0	1
Optic canal	4	40	6	60	0	0	10	100	0.087
Orbital fissure	4	40	6	60	2	20	8	80	0.629
Rostral alar foramen	4	40	6	60	1	10	9	90	0.303
Foramen Ovale	3	30	7	70	1	10	9	90	0.582

**Table 2 animals-14-01643-t002:** Results of the anatomical dissection in ten specimens with each approach. Presence (yes/no) of dye in the structures (intraconal, intramuscular, and interfascial) and nerve staining (yes/no).

Approach	Coronoid				Temporal				*p*-Value
Presence dye/staining	YES	%	NO	%	YES	%	NO	%	
Interfascial	10	100	0	0	8	80	2	20	0.474
Intraconal	0	0	10	100	1	10	9	90	1
Intramuscular	3	30	7	70	4	40	6	60	1
Mandibular nerve	5	50	5	50	5	50	5	50	1
Maxillary nerve	8	80	2	20	4	40	6	60	0.170
Optic nerve	1	10	9	90	0	0	10	100	1
Orbital fissure complex	8	80	2	20	3	30	7	70	0.070

## Data Availability

All the data produced in the study is available in the article.

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
