# Peer review of "Comparison between Ultrasonographic-Guided Temporal and Coronoid Approaches for Trigeminal Nerve Block in Dogs: A Cadaveric Study"

_animals, 2024, doi:10.3390/ani14111643_

Round 1

Reviewer 1 Report

Comments and Suggestions for Authors

Hi,

Many thanks for this great submission. It is fantastic to see someone finally looking into this and I am impressed with thea mount of work that went into putting these results together.

I have a few minor comments below:

L69: instead of V cranial nerve i would write : "the fifth cranial nerve" OR "cranial nerve V"

L71-82: best anatomical description of the trigeminal nerve components I have read.

L140: I feel like there is a distance missing here: "significant difference of two nerve branches stained..." how many mm? 

L238: you probably mean that the approach could be guessed or determined from the CT image evaluation.

Table 1/Fugure 4 and Table 2/Figure 5: are these table and figures not repeating the same information twice?

L262: what's the carium?

L263: withing?

L282: I think there is a discrepancy between your hypothesis/premisses and this conclusion. The approach that is known is the temporal approach and you are hypothesising that the coronoid approach is a superior approach because of higher staining, but the results don't support your hypothesis. So I would either suggest that the initial hypothesis was actually a non-inferiority one (temporal = coronoid) or that you phrase the conclusion differently, saying that the coronoid approach is not superior to the temporal one HOWEVER, it is non-inferior. 

L331: there is actually another reference that mentions the use of the trigeminal block clinically - see below

Thomson, A.E., Rigby, B.E., Geddes, A.T. and Soukup, J.W. (2020) “Excision of Extensive Orbitozygomaticomaxillary Complex Tumors Combining an Intra- and Extraoral Approach With Transpalpebral Orbital Exenteration,” Frontiers in Veterinary Science, 7, p. 569747. Available at: https://doi.org/10.3389/fvets.2020.569747.

L378: "underscoring" - highlighting

Author Response

Dear reviewer,

First of all, the authors would like to thank you for taking the time to read and correct the manuscript. We are sure that with your help, the manuscript will be improved and hopefully we will publish a high-quality article. Thank you for your kind comment, we are very happy to read that you enjoyed it and appreciated our work.  

Please find in the document below each of your questions with our answers. Please notice that your review is the third one, and for this reason, probably many of the changes that you suggest were already applied and are available in the last version of the manuscript.

All the changes you suggested (and the ones suggested by the other two reviewers) are highlighted in red colour in the new version of the manuscript.

Kind regards

Reviewer 2 Report

Comments and Suggestions for Authors

Dear authors,

thank you very much for the submission of your manuscript about the cadaveric study for comparison of 2 approaches for US-guided trigeminal nerve block in dogs. 

This cadaveric study  can be considered as basics for further implementation of clinical studies. The anatomical description and potential complications are of great importance for clinical use. The study is well described with an anatomical description of the relevant structures and further on by appropriate description of the approach.

Some minor comments:

- why has this been choosen as a superior study (coronoid approach better than temporal approach). without other preexisting studies a simple comparison of both approaches would have been appropriate,as also indicated by the title (hypothesis shows a assumed superiority, whereas title does not)

- i am slightly confused by the wording - we call it trigmenal block, but actually do not approach the trigeminal nerve itself, but the location of its main branches. however, as shown in table 2 - eg. the orbital fissure complex (consisting of V1 branches) is only stained in 30% with the temporal approach. so i am not sure, if the term trigeminal block is so good to use.

- i wonder if it would be possible to highlight the sphenoidal complex /target region in the anatomical dissection of the skull (supplementary material 1)

- for figure 2/description of the coronoid approach: the skull is only half visible, also mouth doesn't seem to be open (in comparison to the text). and the US probe seems to be placed more from a caudal direction rather than from dorsal. however, the skull/probe picture is only a snippet and it is difficult to get dimesions/directions.

- results line 233-235 and line 290:how was this observation made. In the description of the preliminary/anatomical study no needle placement was described. So please clarify, whether needle had been placed in this part and if so, before or after dissection or is it based on an assumption looking at the anatomy. It is unclear where this results emerge from.

line 235: what anatomical alterations were presented? was it only related to freezing/unfreezing or potential anatomical variations

please include anatomical structures/descriptions in the relevant CT images (identification of target areas etc) to have it more comprehensive and easier to follow

i don't think a table  and a figure is necessary for the same results, which is also extensively mentioned in the text without any significant finding (relevant to table 1/figure 4 and table 2/figure 5, respectively). i would remove the figures. instead maybe the supplementary CT images could be included somewhere in the main manuscript for better visualisation.

Discussion: even if no difference between both approach, it might be worth mentioning, that the described " trigeminus block" actually does not block all 3 branches of the trigeminal in a fair amount of specismen. this should be mentioned in the discussion and maybe also potential impact on targeted analgesia (eg whether V1, V2 or V3 is relevant for the intended purpose)

- please also comment/discuss on the used volume (as it was reduced compared to another study in an attempt to minimize intracranial spread). does the chosen volume seems appropriate considering that aspect?

Author Response

Dear Reviewer 1,

Thank you very much for your comments. Please find the answers in the document attached. 

Best regards

Reviewer 3 Report

Comments and Suggestions for Authors

I have very much enjoyed reading this manuscript and love that the authors are using ultrasound guidance to try to improve on regional blocks where long needles are passed deep into the pterygopalatine fossa. Please carry on the good work!

I have only a few minor comments.

Simple summary

Line 26: ‘computer tomography’ should be, ‘computed tomography scan’.

Line 27: ‘media’ should be, ‘medium’.

Abstract

Line 39: ‘iodate’ should be (I think), ‘iodinated’.

Introduction

I wonder if mention should be made of the similarities in the needle approaches required for the ‘trigeminal’ block and the more historical blocks of its various branches (maxillary, ophthalmic); and also the retro/peribulbar block).

Line 66: ‘and a shorter’ should be, ‘and shorter’.

Lines 69-71: ‘trigeminal nerve… and it emerges from the skull through the trigeminal canal’  – but the three branches of the trigeminal nerve emerge from the skull via different foramina. The ‘trigeminal canal’ is a boney furrow on the inside of the caudal part of the cranial fossa, near the tympanic bulla. May be this statement could be written as: ‘trigeminal nerve…. and it courses through trigeminal canal (on the interior aspect of the skull), before dividing’.

Line 73: ‘further divides, would be better as, ‘where it further divides’.

Line 75: I understood that the maxillary branch leaves the skull through the foramen rotundum before entering the alar canal?

Line 80: ‘providing’ should be, ‘provides’.

Line 88: ‘commonly’ should be, ‘common’.

Materials and Methods

Line 115: ‘sphenoidal complex’ – does this mean the complex of bones which make up the sphenoid?

Line 128: ‘aspect’ should be, ‘aspects’.

Line 132: ‘alveolar caudal nerve’ should be, ‘caudal alveolar nerve’ (sometimes referred to as the ‘inferior alveolar nerve’).

Line 149: ‘injectate consist of’ should be, ‘injectate consisted of’.

Line 150: ‘iodate’ should be (I think), ‘iodinated’.

Line 194: ‘injection’ should be, ‘injections’.

Lines 195-6: ‘tip of the needle and the target points’ should be, ‘tip of the needle to the target points’.

Lines 212-213: ‘mandibular’ might be better as, ‘mandibular nerve’.

Results

Line 238: ‘sorted out’ would be better as something like, ‘devised’.

Line 245: Fig 3. ‘distance distribution’ should be, ‘distribution distance’.

Line 261: Fig 4. ‘withing’ should be, ‘within’.

Line 262: Fig 4. ‘carium should be, ‘cranium’.

Line 262: Fig 4. ‘withing’ should be, ‘within’.

Line 275/6: Fig 5. ‘withing’ should be, ‘within’.

Line 277: Fig 5. ‘intesfascial’ should be, ‘interfascial’.

Discussion

Line 283: ‘viable alternative’ should be expanded to, ‘viable alternative to the temporal approach’.

I am intrigued, however, that your results, although not statistically significant, showed that contrast medium reached more nerves with the coronoid approach… and that the needle was about 1mm closer to these structures for the coronoid approach too. I wonder if the study had included more samples whether you would have reached statistical significance?
Line 284: ‘its efficacy’ – do you mean the coronoid approach, or the temporal approach – or both?

Line 310: ‘In contrast to our method’ should be, ‘This is in contrast to our method’.

Line 320: ‘tip of the needle over the surface of the maxilla’ might be better written as, ‘tip of the needle onto the surface of the maxilla’?

Line 389: ‘eye oedema’ – does this mean conjunctival oedema, periorbital oedema or eyelid oedema?

Supplementary Materials

Line 418: ‘contrast media’ should be, ‘contrast medium’.

Figures

Fig 3. ‘distance distribution’ should be, ‘distribution distance’.

Fig 4. ‘withing’ should be, ‘within’.

Fig 4. ‘carium should be, ‘cranium’.

Fig 4. ‘withing’ should be, ‘within’.

Fig 5. ‘withing’ should be, ‘within’.

Fig 5. ‘intesfascial’ should be, ‘interfascial’.

Comments on the Quality of English Language

Very well written. I have made a couple of tiny suggestions where words or phrases might be improved.

Author Response

Dear reviewer 2, 

Thank you very much for the comments and corrections, the manuscript improved a lot after applying them. Please find attached a word document with the answer to each of your comments. Note that there are changes in the figures and text corresponding to the other reviewer. 

Kind regards
